# Development of a Digital Case Management Tool for Community Based Inclusive Development Program

**DOI:** 10.3390/ijerph182011000

**Published:** 2021-10-19

**Authors:** Manjula Marella, Donna Koolmees, Chandalin Vongvilay, Bernard Frank, Wesley Pryor, Fleur Smith

**Affiliations:** 1Nossal Institute for Global Health, Melbourne School of Population and Global Health, The University of Melbourne, Melbourne 3010, Australia; wesley.pryor@unimelb.edu.au (W.P.); smith.f@unimelb.edu.au (F.S.); 2World Education Laos, Vientiane 6782, Laos; donna_koolmees@la.worlded.org (D.K.); chandalin_vongvilay@la.worlded.org (C.V.); bernard_franck@la.worlded.org (B.F.)

**Keywords:** disability assessment, support needs, digital data collection, automated analysis, community based inclusive development

## Abstract

Disability inclusive development practices require reliable data to identify people with disabilities, their barriers to participation and support needs. Although several tools are available for measuring different components of disability, it is often difficult for program teams in low resource settings, including lay community workers of community based inclusive development (CBID) programs, to collect and analyze data for program monitoring and evaluation. This paper presents the development of a digital CBID Modular Tool with automated data analysis to support routine case management processes and monitoring of a CBID program in Laos PDR. The tool was developed in different phases involving stakeholder consultations, auditing of existing tools, content development for the different modules for disability assessment and support needs, software development and testing. The tool was developed in a participatory process including people with disabilities. The tool measures needs and support requirements of people with disabilities in health, functioning, economic, education and caregiver support domains, and enables intervention planning. The content included is both context specific and universal as derived from the widely used validated tools. This unique digital CBID Modular Tool can support data collection by lay community workers and support reliable data collection to measure disability inclusion in a development program.

## 1. Introduction

People with disabilities are more likely to experience poverty, poorer health outcomes, and lower education and employment rates compared to people without disabilities [1]. Considering these inequalities, disability is now recognized as a significant development issue [2]. Disability inclusion in development programs aims to promote equal participation of people with disabilities by addressing their social, health, educational and economic barriers. Disability inclusion is an increasingly common focus or requirement in development programs, but there is little evidence to demonstrate its effectiveness. Although efforts are being made internationally by different agencies, data on disability are still lacking from international development program monitoring systems [3]. Recognizing this need, the United Nations’ (UN) Sustainable Development Goals (SDGs) 2015–2030 explicitly mention disability in five of the 17 goals; with Target 17 specifically focusing on disaggregation of data by disability [4]. International funding agencies such as the Australian Department of Foreign Affairs and Trade (DFAT) and the United Kingdom Foreign, Commonwealth & Development Office (FCDO) now require the provision of disability disaggregated data as part of program reporting.

The World Health Organization’s (WHO) International Classification of Functioning, Disability and Health (ICF) defines disability as “an umbrella term for impairments, activity limitations and participation restrictions, referring to the negative aspects of the interaction between an individual (with a health condition) and that individual’s contextual factors (environmental and personal factors)” [5]. Including these various components in one measure is complex. The Washington Group short set on functioning (WG-SS) questions are recommended for collecting data on disability for SDG indicators. These questions focus on the activity limitations component of the ICF covering the most basic functions: seeing, hearing, walking, communicating, remembering and self-care. The WG-SS questions are found to capture the majority of people with disabilities in a culturally sensitive way in different settings [6]. Data on the level of participation of people with disabilities in programs and the contextual factors that have limited/influenced their participation are also critical for monitoring disability inclusion. The Model Disability Survey developed by WHO covers different components of the ICF including participation [7], but the complex analysis required for disability assessment makes its use prohibitive for many development program stakeholders.

The Rapid Assessment of Disability (RAD) survey was developed based on the principles of the UN Convention of the Rights for Persons with Disabilities (CRPD) and ICF frameworks for supporting the design, implementation and evaluation of disability inclusive development activities [8,9]. The RAD survey identifies people with disabilities based on the Washington Group type of questions and measures the impact of disability on inclusion and participation in the community, including the barriers—incorporating different components of disability into four modules of the survey. The RAD survey has been validated in Bangladesh and Fiji [8]. By developing specific additional modules for planning different programs, RAD has been used in an inclusive education program in Fiji [10]; in a participatory research program in the Philippines to improve sexual and reproductive health for women with disabilities [11]; in community-based health programs in India [12]; a program to strengthen access and quality of rehabilitation in Bangladesh [13]; and to inform disability inclusion responses to Tropical Cyclone Pam in Vanuatu [14]. While RAD has been used for program baseline surveys, further research on adapting RAD for monitoring disability inclusion over time and measuring program outcomes is still needed.

While there are several tools available for measuring disability and participation of people with disabilities in the community, it is a challenge for development program managers to adopt them for their programs and analyze the data for reporting. Community-based inclusive development (CBID), which was originally known as community-based rehabilitation, is a community development strategy aiming to maximize opportunities for health, education, livelihood, social life, and community participation for people with disabilities and their families [15]. Although CBID has a history of more than 50 years and has evolved over time, evidence on its effectiveness is limited [16,17,18].

The concept of CBID is multidimensional and there is no single tool that can be used to measure its effectiveness. While there are several approaches proposed for monitoring and evaluation of CBID programs, recent research on the consensus for CBID evaluations identified mixed methods, participatory approaches and a combination of indicators generated locally along with other CBID indictors [19]. Data collection is key for any program evaluation, and in case of CBID programs, data are collected by lay community workers either as part of their case management or using specific tools for evaluations. Further analyzing the data from different dimensions of disability and CBID components is complex for program managers in low resource settings.

Many development programs in low resource settings, including CBID programs, often rely on external technical assistance for monitoring and evaluation. Automated data analysis systems for use at a development program level, particularly in low resource settings, will enable users to engage with data meaningfully and make decisions to support evidence-based disability inclusion without complex data analysis capabilities.

Mobile phone technology is increasingly being used in mHealth as well as in development programming for monitoring and evaluation. Mobile phones, particularly smartphone usage, are increasing globally, with an estimate that half of the global population is using mobile internet [20], and a median of 45% adults in low- and middle-income countries owning a smartphone [21]. Mobile technology is being used for monitoring and evaluating water, sanitation, and hygiene (WASH), microfinance, community-based health, and other development programs in low- and middle-income countries [22,23]. Particularly in the health care sector, evidence suggests that using mobile-based data collection tools has improved the quality of data collection by community health workers compared to paper-based tools, due to the reduced likelihood of errors and data loss [24,25]. Further, real-time data collection assists instantaneous data review, analysis and decision making; and timely management of health issues [25,26]. In the current era of data revolution, given the availability and affordability of mobile technology in low resource settings, developing a mobile technology-based methodology that extends the utility of existing disability tools and validating such a methodology are the logical next steps for improving disability data collection.

### The CBID Demonstration Model in Laos PDR

The United States Agency for International Development (USAID) Okard (ໂອກາດ) Project is a 5-year project aiming to improve and sustain functioning and independent living for people with disabilities in Lao PDR [27]. The Project is funded by USAID and is managed and implemented by the non-government organization World Education Inc. in close collaboration with Humanity and Inclusion and with government and non-government partners. The main components of the Project are health, economic empowerment, stakeholder engagement, and the CBID demonstration model. The CBID demonstration model tests the effectiveness of the interventions in the other components on disability inclusion at community level.

The USAID Okard Project is implementing the CBID program as a demonstration model in the two target districts—Kham District, Xieng Khouang Province and Xayphouthong District, Savannakhet Province. The CBID model encompasses individual case management and community mobilization to remove barriers by utilizing innovative and effective interventions that directly address the health, livelihoods, and social needs of people with disabilities, with a focus on individuals, families, and communities. Two local not-for-profit organizations lead the implementation of the model in close collaboration with district and provincial government partners including the Ministry of Labor and Social Welfare, Ministry of Health, Ministry of Education and Sports and Ministry of Agriculture and Forestry. The CBID teams are lay social workers in each district, comprising one team leader, four CBID facilitators, one income generation activities officer, and one self-help group facilitator. This team directly supports people with disabilities to identify their needs and work with families, communities, local authorities and relevant service providers to meet their support requirements.

For a complex program of this nature, the program partners and stakeholders identified the need for a data collection and analysis tool for case management processes as well as for evaluation of the program. A cloud-based data collection and automated data analysis tool was considered essential for real time data monitoring and reporting on the overall program outputs and outcomes. In this paper, we present the development process of a digital CBID Modular Tool for case management in the USAID Okard’s CBID demonstration model.

## 2. Materials and Methods

The process of developing the digital CBID Modular Tool is shown in Figure 1. The first phase involved consultations with key stakeholders to establish the objectives of the modules and list interventions that are provided through the CBID Demonstration Model to develop a database for individual action plans. The second phase involved content development for the modules and testing. The third phase was digitization and testing the digital tool. The final phase involved training of CBID teams and implementation of the digital tool in the program, which will be reported in a forthcoming paper.

### 2.1. Phase 1. Establishing the Objectives of the Modular Tool

This research is guided by two conceptual frameworks—the UN CRPD [28] and the ICF [5]. Article 31 (Statistics and data collection) of the UN CRPD calls State Parties to undertake research and data collection on disability to “enable them to formulate and implement policies to give effect to the present Convention” and Article 32 (International cooperation) emphasizes the importance of international collaboration for “facilitating and supporting capacity-building, including through the exchange and sharing of information, experiences, training programmers and best practices” [28]. To build evidence on disability-inclusive practices, data on the outcomes of interest, particularly around participation of people with disabilities in the community, is needed in addition to their health and impairment related information. The ICF provides a common framework for understanding disability and a systematic approach to measure the level of functioning and the factors that affect participation [5]. Client-reported outcomes is the approach this project undertook, aiming to give voice to people with disabilities. The ‘Activities and Participation’ and ‘Environmental Factors’ components of the ICF are useful for measuring client-reported outcomes. The approach in this project also aligns with the CRPD’s human rights-based approach focusing on the support needs for people with disabilities, while considering their choice and autonomy to achieve full participation in the community [29].

Consultations were conducted with key stakeholders for defining the objectives of the Modular Tool, the rationale for collecting data and how the data will be used. Key stakeholders included the Okard Project team comprising international and national non-government organizations (World Education, Humanity and Inclusion, Quality of Life Association and the Association for Rural Mobilization and Improvement) implementing the CBID demonstration model, government personnel from relevant ministries, representatives of organizations of people with disabilities, and academic experts. In addition to these consultations, the USAID Okard Project’s results framework was reviewed for the indicators on monitoring and evaluation to ensure the content of the Modular Tool includes data for these indicators.

The list of interventions that can be delivered through the CBID demonstration model was also compiled. This step was undertaken to ensure the content selected for the modules provided relevant information on the individual and household needs, which can be mapped against the interventions. Information on the list of interventions was later used to develop a database for making individualized action plans. The database was designed for automated data analysis of the scores from the Modular Tool data and then to link the scores to relevant interventions.

An audit of existing tools was undertaken to identify the potential content for the modules. Tools were considered if they met the following criteria: developed or validated after the year 2000; used for assessments of needs specific to each focus area of the modules; and used in multiple settings internationally. Where possible, content was identified from the tools that were previously used or tested in the Mekong region to ensure validity and cultural appropriateness of the CBID Modular tool content.

### 2.2. Phase 2. Content Development and Testing

Based on the objectives of each module, the researchers developed a pool of items from the pre-existing tools to be comprehensive in coverage. Each module was then reviewed by the team including the key stakeholders to address why and what each question will provide data on and how they will be linked with the list of interventions mapped. Any content that could not be mapped against the interventions was not included in the modules. The process was iterative and involved focusing on the wording of items, adapting items to the local context, and reviewing it further. Where content was not available from the existing tools, new questions were developed. Careful considerations were made to keep the number of items in each module to a minimum so that the CBID facilitators and the individuals responding to the items do not find the process too arduous. For each item, a response scale was identified based on the previously validated tools and according to the feasibility of translations into the Lao language.

After the content was developed, the modular tool was drafted organizing the items in a sequence for each module and developing skip logic as required. Items in the Modular Tool were structured for children and adults as shown in Figure 2. The modular tool was then translated into Lao and then back translated into English. The paper-based modular tool was further reviewed by the key stakeholders to ensure the flow and appropriateness of the content.

A scoring method was established to support the development of decision trees for each module to identify red flags for unmet needs for each module. This information was then used for developing the database for automated data analysis and to determine the eligibility of the individuals to each type of intervention. Examples of these decision trees are provided later in this paper.

As shown in Figure 1, initial testing of the paper-based modular tool was undertaken on a network of family and friends including people with disabilities to establish face validity. This level of testing focused on the clarity of the questions, and response options, how well the respondents understood the meaning of the questions and response options, and if there were any items that were considered not relevant for their situation. The content was then further refined based on this testing.

### 2.3. Phase 3. Digitization and Testing

The modular tool content was digitized using KoBoCollect, an application from the KoBoToolbox, which is an open-source platform for data collection and analysis. Data can be collected using smartphones/tablets online or offline. Data are temporarily stored on the phone when offline and uploaded into the digital cloud database when the internet connection is available. Data from the modular tool are then linked to a customized case management database designed using cloud-based Amazon Web Services. This database automatically analyses the data for identifying the unmet needs in each module based on the algorithm developed using the decision trees. A list of interventions is created in the database that can be selected and linked for the identified unmet needs to prepare an individual action plan.

The first version of the digital modular tool was initially tested with the known networks of people with disabilities. This testing focused on the language, response options, skip logic and the overall structure of the modules. After refining the errors, the second version of the Modular Tool was further tested with purposively selected people with disabilities with different levels of functioning difficulties, people without functioning difficulties, and those who use assistive products from a sample of families in the target areas of the CBID project. This level of testing identified errors with understanding some questions, and translations. The technical team consulted again to refine the content further and a pilot version was developed for training the CBID team and implementation of the project.

## 3. Results

### 3.1. Objectives of the Modular Tool and Its Rationale

The key objective of the digital modular tool was to generate data on the unmet needs and support requirements for people with disabilities and their households to make evidence-based decisions on individual case management action plans and monitor the progress.

It was agreed that the modular tool will comprise of these modules: Demographics, Education, Economic participation, Function and Assistive products, Health conditions, Mental health, Caregiver, Access and utilization of health services, and Wellbeing.

The information from the modular tool data will be used to plan appropriate interventions including referrals for assistive products, health, rehabilitation, psychosocial support and education services, and economic opportunities. Data collected at baseline will be compared to data collected at the time of discharge to monitor changes to the unmet needs and measure outcomes on the different modules. These data collected over time will be regularly used to generate reports on the indicators in the USAID Okard Project results framework. Information on the needs and trends within the target population will be used for refining the CBID demonstration model. Further, the data will be used for strengthening systems in the country by providing support to government partners to use the data and lessons learned for planning and implementation of disability inclusion policies and strategies (Disability Law, Strategy and Action Plan, and National Rehabilitation Strategy).

### 3.2. Content of the Modular Tool

Module 1 on Demographics, Education and Economic participation measures the socioeconomic status of the households, and work satisfaction and access to education and vocational training opportunities for people with disabilities. The content on the individual and household demographics and economic participation was derived from the RAD baseline survey that was undertaken for the CBID demonstration model [30], economic assessments of households undertaken by civil society organizations implementing livelihood projects, the Laos census [31] and Social Indicator Survey [32], and WHO CBR indicators [33].

Household demographics and economic status were measured using items related to the characteristics and make of the house, the number of individuals living in the household, food consumption, ownership of livestock, working situation of adult members of the household, debt, and supports and benefits received. At the individual level, those who are identified as having a disability are questioned on general demographic information including age, sex, ethnicity, language, marital status, and education. Items on economic participation at the individual level include the current work situation, willingness to work and undertake vocational training and related barriers, whether they own a bank account and the frequency of its use, and whether they have financial management plans for their income. These individual level questions are asked to the caregivers of children with disabilities. Education related questions for all school aged children in the household include items on school enrolment and attendance, and barriers to schooling.

Module 2 on Function and Assistive Products measures difficulties in functioning, i.e., activity limitations and participation restrictions with or without assistance, and information on assistive product use, type, source, and benefits. This module uses standard question sets by WG, including the extended set of questions for functioning [34] and the WG/UNICEF Child Functioning Module [35]. Additional questions on each domain of functioning were derived from the World Health Organization Disability Assessment Schedule (WHODAS) 2.0 [36], and WHO Assistive Technology Assessment (ATA) toolkit [37]. The domains of functioning included in the module are seeing, hearing, communication, mobility, moving arms and hands, self-care, cognition, participation in activities of daily living, behavior, and play. As shown in Figure 2, question sets are different for 2–4 y, 5–17 y, and 18 y and over.

Module 3 on Health conditions measures the need for interventions related to the health and impairments for people with disabilities. Items included in this module are mostly customized according to the health conditions most frequently recorded and treated at district and provincial hospitals and rehabilitation centers to inform the type of health-related interventions facilitated through the CBID demonstration model. The items included at the individual level ask for general health rating, health conditions that need immediate attention and those that limit activities of daily living, causes, use of medications, whether they have enough supply of medication and any treatments currently taken/advised.

Module 4 on Mental health measures the need for psychosocial support for people with disabilities. This module comprises of Adapted Pediatric Symptom Checklist (PSC) for eight to 10 years old [38,39], Patient Health Questionnaire—Adolescents (PHQ-A) for 11 to 17 years old [40], and Patient Health Questionnaire (PHQ-9) [41], and Primary Care Post Traumatic Stress Disorder screen for DSM-5 (PC-PTSD) for 18 years and older [42]. Standard recommended scoring methods are used to determine the symptoms of stress, anxiety, depressions and/or post-traumatic stress disorder for the individuals with disabilities and their caregivers.

Module 5 on Caregiver measures the support needs and requirements of the main caregiver. Items in this module are derived from the Adult Carer Quality of Life (AC-QoL) questionnaire [43] to ask different experiences of caregiving and support needed for caregiving. In addition, this module includes questions on general health, medication, mental health (using PHQ-9 and PC-PTSD questionnaires) and subjective wellbeing (based on Satisfaction with Life Scale—SWLS) [43].

Module 6 measures the access to and utilization of health services for people with disabilities including barriers. The items ask about the health seeking behaviors to identify what actions people with disabilities took when they were ill and what services they chose to access, and access to rehabilitation and mental health services. Barriers related to accessing and utilizing these services are also asked. The items for this module were derived from RAD and other surveys undertaken on health care seeking behavior in the region.

Module 7 measured subjective wellbeing of people with disabilities, which was one of the monitoring and evaluation indicators required for reporting on the USAID Okard Project’s results framework. Wellbeing was defined as the general sense of satisfaction with life by the individuals and was measured using the SWLS for 15 years and older [43], and SWLS adapted for children (SWLS-C) for 9–14 years [44]. These questions were asked directly to the individual and not by a proxy. This module was not asked if the respondent could not respond to these questions by themselves.

### 3.3. Development of the Database for Automated Analysis

The question sets in each module were reviewed to develop a scoring system and threshold score levels to flag the need for support. Decision trees, which are flow charts to determine the action plan for each trigger response, were created to support the development of algorithms for automated data analysis and linking the relevant interventions for developing an action plan. A total of 28 decision trees were developed for all question sets varying in complexity using the thresholds determined for trigger responses. Two specific examples of decision trees are discussed in this paper.

Example 1: The self-care domain in Module 2 has six questions for both 5–17 y, and 18 y and over question sets (Table 1). The questions ask about difficulties regarding washing, dressing, grooming, toileting, eating and drinking, and information on assistive products used. The thresholds for identifying needs are highlighted in Table 1. Two decision trees are developed separately for the first four questions (Figure 3a) and eating and drinking (Figure 3b) questions because interventions are different for each of these sets that can be provided at referral centers and directly by the CBID facilitators. When an individual responds as having a lot of difficulty or cannot do at all for at least one of the six activities, and has challenges accessing or using assistive products for self-care, it is considered that the individual needs support. A step-by-step action plan for referral to rehabilitation services and assistive product assessment is triggered for that individual. Referral services existing in each of the provinces are already mapped and essential linkages are made in coordination with the Center for Medical Rehabilitation of the Ministry of Health supported by the USAID Okard Project. The database prompts the CBID facilitator to discuss options for supporting self-care needs with the individual and their families.

Example 2: Module 6 (Table 2), on access to and utilization of health care, has two questions on healthcare seeking behavior in the past when the individual was ill, and three questions each on accessing health services, rehabilitation and assistive product services and mental health and psychosocial support (MHPSS) services, asking for the level of access to services and barriers to access. The thresholds for responses to these questions highlighted in Table 2 indicate a trigger for the CBID facilitator to consider interventions on health literacy or other support to improve access to services (Figure 4). The algorithm developed based on the decision trees for the automated database is as follows:

IF:I6_6a.3aE_T, I6_6a.4aE_T and/or I6_6a.5aE_T = 0, 1 or 89or I6_6a.2E*_T = 0,1,2,3, 6 or 7

THEN consider relevant interventions.

**Table 2 ijerph-18-11000-t002:** Module 6, Access to and utilization of health services.

Number	Questions	Response Categories	Code	Comments and Skips
I6_6a.1E*_Q	Thinking of the time you were ill last time, what was your first/immediate point of action? (*Do not read options. Select one option)*	No action	0	Go to I6_6a.2E*_T
		Traditional method by self	1	
		Western medicine by self	2	
		Pharmacy/dispensary	3	
		Public health facilities	4	
		Private health clinic/ practitioner	5	
		Community health volunteer in the village	6	
		Traditional healer (use local term)	7	
		Other	8	
		Specify other	(Text)	
I6_6a.2E*_T	Thinking about the illness you had, what did you do when the illness did not go away or worsened? (*Do not read options. Select one option)*	** *No action* **	** *0* **	Go to I6_6a.3aE_T
		** *Traditional method by* ** ** *self* **	** *1* **	
		** *Western medicine by* ** ** *self* **	** *2* **	
		** *Pharmacy/dispensary* **	** *3* **	
		Public health facilities	4	
		Private health clinic/ practitioner	5	
		** *Community Health* ** ** *volunteer in the village* **	** *6* **	
		** *Traditional healer (use* ** ** *local term)* **	** *7* **	
		Other	8	
		Specify other	(Text)	
I6_6a.3aE_T	In the last 3 months, to what extent have you been able to access health services? (*Read all options and select one*)	As much as needed	3	Go to I6_6a.4aE_T
		Most times	2	Go to I6_6a.4aE_T
		** *Sometimes* **	** *1* **	Go to I6_6a.3bE*_Q
		** *Not at all* **	** *0* **	
		Have not needed	4	Go to I6_6a.4aE_T
		** *Don’t know* **	** *89* **	Go to I6_6a.4aE_T
I6_6a.3bE*_Q	What are the reasons/difficulties for not being able to access health services as much as you needed? *(Do not read options, select all applicable options)*	Lack of information/do not know where to go	1	Go to I6_6a.3cE*_Q IF more than one response selected in I6_6a.3bE*_Q
		No services/facilities	2	
		Too far	3	
		No transport available	4	
		Transport not accessible	5	
		Could not afford the cost of the visit	6	
		Could not afford the cost of transport	7	
		Nobody to accompany me	8	
		Do not like the attitudes of the staff at the facility	9	
		Was previously badly treated	10	
		Tried but denied health care	11	
		Did not think sick enough	12	
		Other	13	
		Specify other	(text)	
I6_6a.3cE*_Q	Among the reasons you have listed which one has limited your access to health services the most? *(Do not read options. Select one option)*	Lack of information/do not know where to go	1	When asking the question only list options selected in I6_6a.3bE*_Q Go to I6_6a.4aE_T
		No services/facilities	2	
		Too far	3	
		No transport available	4	
		Transport not accessible	5	
		Could not afford the cost of the visit	6	
		Could not afford the cost of transport	7	
		Nobody to accompany me	8	
		Do not like the attitudes of the staff at the facility	9	
		Was previously badly treated	10	
		Tried but denied health care	11	
		Did not think sick enough	12	
		Other	13	
		Specify other	(text)	
I6_6a.4aE_T	In the last 3 months, to what extent have you been able to access rehabilitation services and assistive products? (*Read all options and select one*)	As much as needed	3	Go to I6_6a.5aE_T
		Most times	2	Go to I6_6a.5aE_T
		** *Sometimes* **	** *1* **	Go to I6_6a.4bE*_Q
		** *Not at all* **	** *0* **	
		Have not needed	4	Go to I6_6a.5aE_T
		** *Don’t know* **	** *89* **	Go to I6_6a.5aE_T
I6_6a.4bE*_Q	What are the reasons/difficulties for not being able to access rehabilitation services and assistive products as much as you needed? *(Do not read options and select all applicable options)*	Lack of information/do not know where to go	1	Go to I6_6a.4cE*_Q IF more than one response selected in I6_6a.4bE*_Q
		No services/facilities	2	
		Too far	3	
		No transport available	4	
		Transport not accessible	5	
		Could not afford the cost of the visit	6	
		Could not afford the cost of transport	7	
		Nobody to accompany me	8	
		Do not like the attitudes of the staff at the facility	9	
		Was previously badly treated	10	
		Tried but denied health care	11	
		Did not think sick enough	12	
		Other	13	
		Specify other	(text)	
I6_6a.4cE*_Q	Among the reasons you have listed which one has limited your access to rehabilitation services and assistive products the most? *(Do not read options. Select one option)*	Lack of information/do not know where to go	1	When asking the question only list options selected in I6_6a.4bE*_Q Go to I6_6a.5aE_T
		No services/facilities	2	
		Too far	3	
		No transport available	4	
		Transport not accessible	5	
		Could not afford the cost of the visit	6	
		Could not afford the cost of transport	7	
		Nobody to accompany me	8	
		Do not like the attitudes of the staff at the facility	9	
		Was previously badly treated	10	
		Tried but denied health care	11	
		Did not think sick enough	12	
		Other	13	
		Specify other	(text)	
I6_6a.5aE_T	In the last 3 months, to what extent have you been able to access services (MHPSS *) to help you with stress, anxiety, depression such as, a doctor with specialist skills in mental health or community health worker, peer to peer support or a social club (*Read all options and select one*)	As much as needed	3	Go to I7_1a.1E_Q
		Most times	2	Go to I7_1a.1E_Q
		** *Sometimes* **	** *1* **	Go to I6_6a.5bE*_Q
		** *Not at all* **	** *0* **	
		Have not needed	4	Go to I7_1a.1E_Q
		** *Don’t know* **	** *89* **	Go to I7_1a.1E_Q
I6_6a.5bE*_Q	What are the reasons/difficulties for not being able to access to services (MHPSS) to help you with stress, anxiety, depressions as much as you needed? (*Do not read options. Select all applicable options)*	Lack of information/do not know where to go	1	Go to I6_6a.5cE*_Q IF more than one response selected in I6_6a.5bE*_Q
		No services/facilities	2	
		Too far	3	
		No transport available	4	
		Transport not accessible	5	
		Could not afford the cost of the visit	6	
		Could not afford the cost of transport	7	
		Nobody to accompany me	8	
		Do not like the attitudes of the staff at the facility	9	
		Was previously badly treated	10	
		Tried but denied health care	11	
		Did not think sick enough	12	
		Other	13	
		Specify other	(text)	
I6_6a.5cE*_Q	Among the reasons you have listed which one has limited your access to services (MHPSS) when you have felt stressed, anxious, or depressed the most? *(Do not read options. Select one option)*	Lack of information/do not know where to go	1	When asking the question only list options selected in I6_6a.5bE*_Q Go to I7_1a.1E_Q
		No services/facilities	2	
		Too far	3	
		No transport available	4	
		Transport not accessible	5	
		Could not afford the cost of the visit	6	
		Could not afford the cost of transport	7	
		Nobody to accompany me	8	
		Do not like the attitudes of the staff at the facility	9	
		Was previously badly treated	10	
		Tried but denied health care	11	
		Did not think sick enough	12	
		Other	13	
		Specify other	(text)	

Note: Text in bold and italics is the threshold for identifying needs. * MHPSS—Mental Health and Psychosocial Support.

**Figure 4 ijerph-18-11000-f004:**
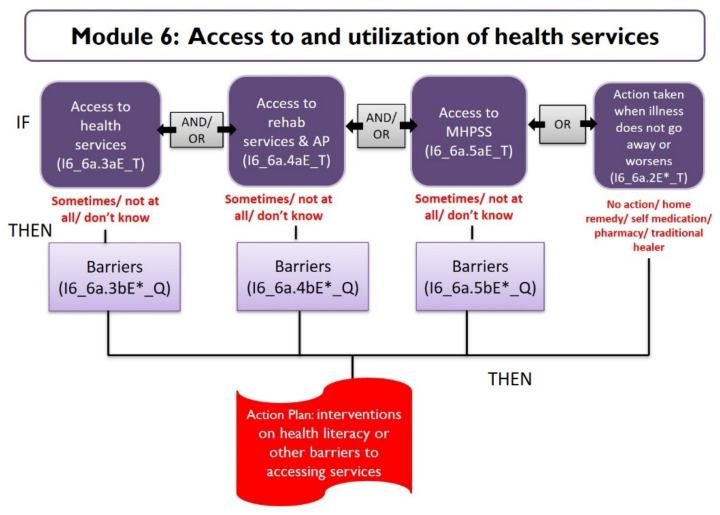
Decision tree for Module 6.

### 3.4. Pilot Module

Testing of the digital tool with the purposively selected sample did not suggest major structural and content changes. Modifications were required for some translations and contextualization of some questions in Lao language, and they were addressed in discussion with the technical team. The revised version of the digital tool was tested again before being released as a pilot version for training of the CBID team. This version took approximately 45 min to 90 min to administer depending on the complexity of needs and including breaks when needed. CBID facilitators can collect data from the families over several shorter sessions depending on the time and availability of the families. However, most families so far have preferred to complete the needs assessment in a single appointment. The data uploaded from the field is regularly validated at the backend by the team leader.

The data are updated when the needs are changed at the individual and family level due to unexpected circumstances, such as the deterioration of existing health condition or developing a new health condition, financial shocks or out-of-pocket expenses, disasters and loss of assets and other reasons. A protocol is created to ensure false data entry is prevented when an update is required. The tool is re-administered when all actions in the action plan have a ‘completed’ status indicating potential for discharge. Results from the re-administration of the modular tool are shared and discussed with families and individuals to come to an agreement that their needs are met to decide whether to discharge or extend the case management.

## 4. Discussion

The CBID modular tool is a comprehensive disability and needs assessment tool specifically developed for implementing health, rehabilitation, economic, education and caregiver support interventions in the community and reporting on the indicators for the USAID Okard Project. It captures information on health, functioning and environmental domains of disability assessment including support needs, aligning with UNCRPD. People with disabilities and their families were involved throughout the process of development and implementation of the modular tool starting from the conceptual phase, through to design, development of question sets and software, training of CBID facilitators, and its implementation. Data from the modular tool facilitates a person-centered approach to the CBID case management process.

The backend database automatically identifies needs for support in each module and develops individualized action plans for interventions. This automated scoring method is anticipated to make it easier for CBID teams in the field, facilitating discussions with persons with disabilities and their families to make informed decisions on interventions. The development and pilot testing of the tool underwent a rigorous process of selection and development of items and decision trees for each module, translations, digitization, and pre-testing.

The CBID modular tool is an innovative technology in the CBID space for supporting data collection, creating action plans and generating evidence on the effectiveness of the CBID demonstration model. The tool is currently being implemented in the field for case management needs assessment as part of the USAID Okard Project, where baseline and follow-up interviews are compared to measure changes to functioning, support needs and environmental and contextual factors overtime after providing interventions.

In an expert consensus study undertaken by Grandisson et al., a panel of experts agreed that the evaluation processes in the CBID field should be inclusive, participatory, empowering and respectful of local cultures and languages [19]. The development of the CBID modular tool was developed in a participatory process involving academics, project managers, CBID team, and key government and non-government stakeholders. The data collected through the digital tool, along with the automatically generated outputs simplifies the monitoring and evaluating processes for the CBID program for lay CBID team members and other relevant stakeholders including the program managers, requiring limited technical skills for data analysis. These simplified processes are expected to support local ownership of the evaluation process and promote a sustainable CBID model beyond the life of the project. The outputs also facilitate discussions with families to better engage them in person-centered care.

The experts in the study undertaken by Grandisson et al. also recommended that data collection tools should be both context-specific and universal [19]. The questions included in the modular tool were carefully identified from the tools that are widely used internationally such as the WHODAS 2.0, PHQ-9, PC-PTSD and AC-QOL to ensure international comparability. These questionnaires were also selected on the basis that they were previously used or tested in the Lao or neighboring countries which are culturally similar. Some questions were also custom developed to meet with the specific objectives of the CBID demonstration model and the USAID Okard Project.

There are some limitations to be considered for this paper. The review of literature was not systematic and not exhaustive. The literature that was sufficient to inform the development of the modular tool was reviewed. This paper only reports the development process of the modular tool and further testing of tool’s useability, feasibility, and effectiveness as part of the case management process in the CBID demonstration model and the validity and reliability of the question sets used still need to be studied. The research team is currently undertaking further research to address some of these limitations and will be reporting in future publications.

## 5. Conclusions

The CBID modular tool is a unique tool comprising of data collection and an automated data analysis system facilitating needs assessment of people with disabilities for use by lay CBID facilitators, and to regularly monitor and evaluate the outputs and outcomes of the CBID demonstration model implemented in Laos PDR. This tool was developed in a participatory process and includes culturally appropriate questions derived from internationally validated and regionally used tools. It is anticipated that data from the CBID modular tool will support evidence-based practices within the CBID demonstration model as well as inform the broader rehabilitation agenda for government partners and other key stakeholders in the country for planning and implementation of disability inclusion policies and strategies. The CBID modular tool will enable better disability data collection, leading to much richer and deeper understandings of disability inclusion than many programs in low resource settings that are currently able to provide. Improved knowledge on effective practices will support global agendas on disability inclusive development for reducing inequalities and enhancing participation of people with disabilities in society.

## Figures and Tables

**Figure 1 ijerph-18-11000-f001:**
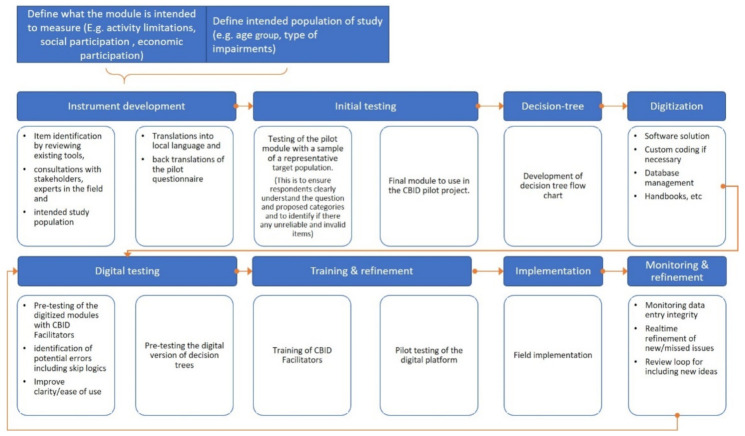
Development process for the modular tool.

**Figure 2 ijerph-18-11000-f002:**
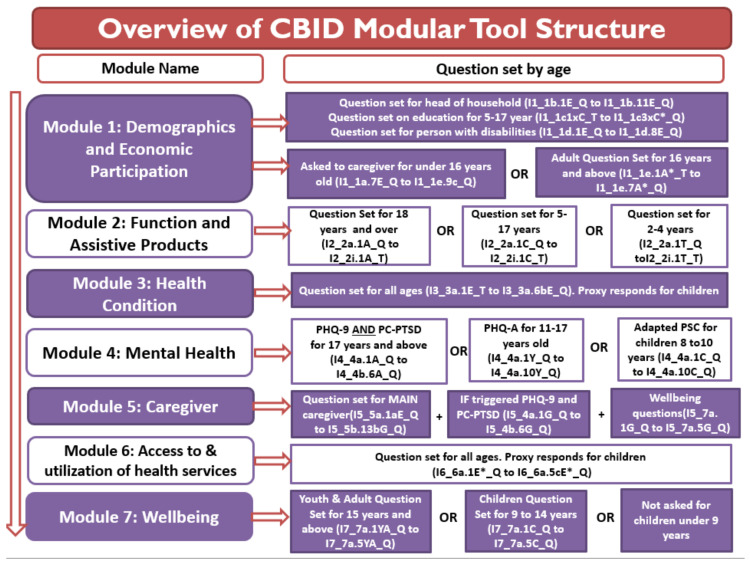
CBID modular tool structure.

**Figure 3 ijerph-18-11000-f003:**
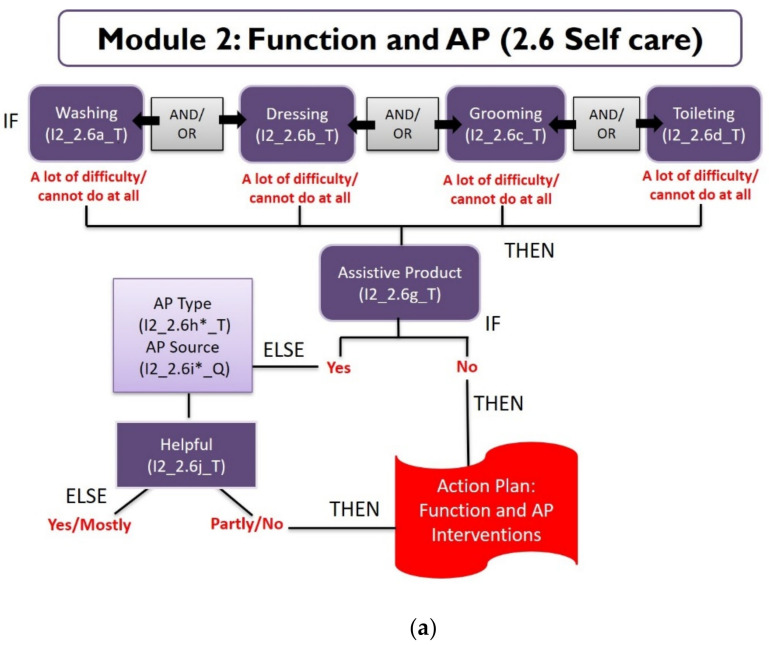
(**a**). Decision tree for washing, dressing, grooming and toileting in the self-care domain. (**b**). Decision tree for eating and drinking in the self-care domain.

**Table 1 ijerph-18-11000-t001:** Questions on self-care domain in Module 2, Function and Assistive Products.

Number	Questions	Response Categories	Code	Comments and Skips
2.6a	Do you have difficulty washing your whole body? *(read all options and select one)*	No difficulty	0	Go to 2.6b
		Some difficulty	1	
		** *A lot of difficulty* **	** *2* **	
		** *Cannot do at all* **	** *3* **	
		Not applicable	99	
2.6b	Do you have difficulty getting dressed? *(read all options and select one)*	No difficulty	0	Go to 2.6c
		Some difficulty	1	
		** *A lot of difficulty* **	** *2* **	
		** *Cannot do at all* **	** *3* **	
		Not applicable	99	
2.6c	Do you have difficulty grooming (e.g., brushing hair, shaving, cutting finger/toe nails and cleaning teeth)? *(read all options and select one)*	No difficulty	0	Go to 2.6d
		Some difficulty	1	
		** *A lot of difficulty* **	** *2* **	
		** *Cannot do at all* **	** *3* **	
		Not applicable	99	

2.6d	Do you have difficulty using the toilet? *(read all options and select one)*	No difficulty	0	Go to 2.6e
		Some difficulty	1	
		** *A lot of difficulty* **	** *2* **	
		** *Cannot do at all* **	** *3* **	
		Not applicable	99	
2.6e	Do you have difficulty eating/feeding yourself? *(read all options and select one)*	No difficulty	0	Go to 2.6e
		Some difficulty	1	
		** *A lot of difficulty* **	** *2* **	
		** *Cannot do at all* **	** *3* **	
		Not applicable	99	
2.6f	Do you have difficulty drinking by yourself? *(read all options and select one)*	No difficulty	0	Go to 2.6e
		Some difficulty	1	
		** *A lot of difficulty* **	** *2* **	
		** *Cannot do at all* **	** *3* **	
		Not applicable	99	
2.6g	Do you use any assistive products to help you eat; for toileting or washing; to dress by yourself or to manage other daily activities?	Yes	1	Go to 2.6h
		** *No* **	** *0* **	Go to 2.7a
2.6h	Which assistive products do you use? (select all applicable options)	Orthoses, upper limb	1	Q2.6i
		Prostheses, upper limb	2	
		Hand rails/grab bars	3	
		Incontinence products, absorbent	4	
		Chairs for shower/bath/toilet	5	
		Adapted cutlery	6	
		Adapted cooking tools	7	
		Transfer board	8	
		Other—device not listed here	9	
		Please specify other	(text)	
2.6i	Where did you get your *product [insert name of the selected product in Q2.6h]*? *read all options and select one)*	Rehabilitation center	1	Go to Q2.6j Repeat Q2.6i and 2.6j IF Q2.6h is more than one assistive product
		Public health facility	2	
		Private health facility	3	
		Local/private market	4	
		Pharmacist	5	
		Made by local handyman	6	
		NGO/Charity	7	
		Purchased outside Laos	8	
		Made by self	9	
		Made by family member/friend	10	
		Other	11	
		Please specify other	(text)	
2.6j	Does the *assistive product* you use help you eat; for toileting or washing; to dress by yourself or to manage other daily activities? *(read all options and select one)*	Yes	1	Go to Q2.7a
		Mostly	2	
		** *Partly/To an extent* **	** *3* **	
		** *No* **	** *0* **	

Note: Text in bold and italics is the threshold for identifying needs.

## Data Availability

Not applicable.

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
