# Peer review of "Development of a Digital Case Management Tool for Community Based Inclusive Development Program"

_ijerph, 2021, doi:10.3390/ijerph182011000_

Round 1
Reviewer 1 Report
The following sentence (line 102-104) requires a reference (or reformulation) as this seem to be a gross generalisation which possibly true for Lao PDR but certainly not for all Asian, let alone African LMICs.
Abbreviation MHPSS is not explained
Contrast Figure 1 should be improved.
Table 2 is a long complex table: may need a better lay-out possibly in landscape? Some edits needed in the table too. The note in line 393 (and 317 for Table 1) should in my view be added at a more strategic place (maybe as an intro to the tables or as a clear footnote?)
In the Figues you use the 'IF'... (great) but why not stating then also the 'THEN' (seems to me useful)
The short reference to the evolution from CBR into CBID is a reductive but I understand the reason to keep it like you formulate this...
Author Response
The following sentence (line 102-104) requires a reference (or reformulation) as this seem to be a gross generalisation which possibly true for Lao PDR but certainly not for all Asian, let alone African LMICs.
Response: Thank you, the statement is now updated with relevant references.
“Mobile phones, particularly smartphone usage is increasing globally with an estimate that half of the global population using mobile internet [20] and a median of 45% adults in low- and middle-income countries owning a smartphone [21].”
Abbreviation MHPSS is not explained
Contrast Figure 1 should be improved.
Response: This is revised.
Table 2 is a long complex table: may need a better lay-out possibly in landscape? Some edits needed in the table too. The note in line 393 (and 317 for Table 1) should in my view be added at a more strategic place (maybe as an intro to the tables or as a clear footnote?)
Response: Table 2 is an example of the complex nature of some modules and how an action plan is generated. Table 2 is formatted and a footnote to indicate the highlights for the thresholds is included in the main text and in the tables.
In the Figues you use the 'IF'... (great) but why not stating then also the 'THEN' (seems to me useful)
Reference: Figures are updated now.
The short reference to the evolution from CBR into CBID is a reductive but I understand the reason to keep it like you formulate this
Response: We tried to provide a brief introduction to CBID in the introduction to provide context for readers and kept it brief to manage the length of the manuscript.
Reviewer 2 Report
ABSTRACT
The objective of the study is well defined and identified both in the abstract and in the introduction.
Good approach to the definition of disability, approached from the difficulties and. Limitations in activities of daily living, such as moving, washing, walking, talking, etc., following the definition of the World Health Organization, with the CIF.
INTRODUCTION:
The subject investigated is of growing scientific and social interest. Research is up to date.
MATERIALS, METHODS and RESULTS:
I would like you to better explain the testing mechanism of the tool, commented on line 224, I don't really understand what it means.
Figure 2 shows the structure of the questions and is differentiated by age group, perhaps it should also have been done by sex, at least older people, who have different pathologies.
Discussion and conclusion:
The conclusions are clear, but I would like to see some more conclusions, in relation to public policy action.
Author Response
Comment 1: I would like you to better explain the testing mechanism of the tool, commented on line 224, I don't really understand what it means.
Response: Thank you for your positive feedback. We have now clarified the initial testing of the paper-based Modular Tool content as below.
“As shown in Figure 1, initial testing of the paper-based Modular Tool was undertaken on a network of family and friends including people with disabilities to establish face validity. This level of testing focused on the clarity of the questions, and response options, how well the respondents understood the meaning of the questions and response options, and if there were any items that were considered not relevant for their situation. The content was then further refined based on this testing.”
Comment 2: Figure 2 shows the structure of the questions and is differentiated by age group, perhaps it should also have been done by sex, at least older people, who have different pathologies.
Response: The questions are not differentiated by sex or different pathologies. There are different question sets for different age groups because some of the functioning questions for adults are not applicable for children. We used pre-existing tools developed for different age groups. As the reviewer noted in their comment above, the Modular Tool content only focuses on the activity limitations and participation restrictions for disability and needs assessments. While data on health conditions/impairments are collected, they are mainly to facilitate appropriate health and rehabilitation referrals are made.
Comment 3: The conclusions are clear, but I would like to see some more conclusions, in relation to public policy action.
Response: We included a statement on the potential use of the Modular Tool data to inform country planning and implementation of the disability inclusion policies and strategies.
Reviewer 3 Report
This paper presents the development of a digital community based inclusive development modular tool helpful for data analysis.
The paper is well written and I really appreciated the quality of the proposed module. However, a pilot study with a reliability analysis of the respondents’ answer would reinforce the results.
Author Response
This paper presents the development of a digital community based inclusive development modular tool helpful for data analysis.
The paper is well written and I really appreciated the quality of the proposed module. However, a pilot study with a reliability analysis of the respondents’ answer would reinforce the results.
Response: Thank you for your positive feedback. We have included a statement on the reliability testing of the Modular Tool under limitations.